# Sex trafficking vulnerabilities in context: An analysis of 1,264 case files of adult survivors of commercial sexual exploitation

**Courtney Furlong** [ORCID]*, **Ben Hinnant**

Department of Human Development and Family Science, Auburn University, Auburn, AL, United States of America

* furloca@auburn.edu

**Data Availability Statement:** This manuscript's data are not publicly available due to restrictions imposed by Frontline Response International, Inc. These restrictions are intended to protect client

## Abstract

### Purpose

Commercial sexual exploitation occurs when anything of value is given in exchange for a sex act. Sex trafficking involves the commercial sexual exploitation of individuals by means of force, fraud, or coercion. Due to the illegal nature of commercial sexual exploitation, there is a profound dearth in the literature. To develop a deeper understanding of the experiences of adult survivors of commercial sexual exploitation, investigators analyzed 1,264 unique case files collected between 2011 and 2021.

### Methods

Key predictors included mental health diagnoses, childhood sexual abuse, and educational achievement, while relevant outcomes included age of entry into sexual exploitation, length of exploitation, number of arrests, cycling into and out of commercial sexual exploitation, and program placement outcomes. Regression analyses (e.g., linear, binomial, or zero-inflated Poisson) were conducted.

### Results

Results suggest that educational achievement is a potential protective factor against exploitation. Higher number of arrest and higher number of children had a bidirectional relationship with longer experiences of exploitation. Further, diagnoses of bipolar disorder and neurodevelopmental disorders were related to higher rates of cycling (i.e., repeated attempts to exit exploitation), and neurodevelopmental disorders and schizophrenia spectrum disorders were related to poorer placement outcomes.

### Conclusions

The findings provide a more authentic portrait of contextual influences on commercial sexual exploitation across a lifespan, informing services, interventions, and policy and supporting survivors in their promising futures.

confidentiality. However, the data underlying the results presented in the study are available from Frontline Response International, Inc. at info@frontlineresponse.org. Data may be provided to eligible individuals upon reasonable request and with the completion of required prerequisites, such as a Data Use Agreement.

**Funding:** The author(s) received no specific funding for this work.

**Competing interests:** The authors have declared that no competing interests exist.

# Introduction

Commercial sexual exploitation (CSE) occurs when anything of value (i.e., money, drugs, shelter, food, clothing, etc.) is given in exchange for a sex act (i.e., intercourse, stripping, lap dancing, pole dancing, erotic massage, escort services, pornography, prostitution, etc.). Sex trafficking is a complex and multifaceted phenomenon that involves the CSE of individuals by means of force, fraud, or coercion [1]. Individuals who are under the age of 18 who are induced to perform a sex act(s) are considered victims of sex trafficking without the condition of force, fraud, or coercion under U.S. federal law. Poverty, family violence, neglect, academic failure, time in the foster care system, and a history of childhood sexual abuse create vulnerabilities for CSE [2–9].

CSE may include experiencing rape, assault, sexually transmitted illnesses, mental health disorders, and murder [3, 10, 11]. Negative health outcomes for individuals who have experienced sex trafficking result from physical violence, psychological disorders due to trauma, addiction, unsafe and violent sex acts, unsanitary living or working conditions, and restrictions in access to healthcare [12]. According to a study conducted by Farley et al., [3] individuals in prostitution in the United States reported that 82% had been physically assaulted, 73% had been raped, 59% had been raped more than five times, and 78% had been threatened with a weapon. Individuals in prostitution in the United States have a standard mortality rate 4 points higher than the general American public (5.9 compared to 1.9) [10]. The leading causes of death for individuals in prostitution were homicide (19%), drug-related causes (18%), accidents (12%), alcohol-related causes (9%), and HIV/AIDS (8%). According to Potterat et al., [10] women in prostitution are 18 times more likely to be murdered while in prostitution than their un-exploited peers. Individuals who have experienced CSE also encounter unique barriers in regard to career, education, relationships, parenting, housing, and economic stability [3].

The present study is the first to investigate the secondary dataset of 1,264 unique case files of adult victims of CSE collected between 2011 and 2021 by a temporary safe home program located in Atlanta, GA. To develop a better understanding of the experiences of survivors as well as identify indicators that may serve to support survivors as they exit exploitation, various potential predictors of CSE, like mental health diagnoses, childhood sexual abuse, and educational achievement, along with relevant outcomes, such as age of entry into CSE, length of exploitation, and placement outcomes, were considered.

## Literature review

### Commercial sexual exploitation

The language of CSE is varied, including terms like "sex trafficking," "prostitution," "sex work," "domestic minor sex trafficking," and "modern day slavery." It is estimated that 84% of individuals in CSE are under third-party control [13]. Reflecting program eligibility requirements of the safe home program (i.e., a history of CSE), the authors of this study will use the broader term "CSE" throughout the investigation, though other terms may be utilized to reflect terminology and their associated definitions used in previous studies. The authors are choosing to use the terms "entry" and "exit," which are intended to be neutral terms acknowledging the reality of victimization along with the agency of survivors. Further, the term "survivor" (rather than victim) will be used to describe study participants to promote strengths-based language and in consideration that participants were not experiencing exploitation at the time their data were collected.

### Dearth in the literature

Research into CSE is limited due to the nature of the crime and challenges researchers have accessing individuals who have experienced exploitation. McCoy [14] reviewed 5 studies that

surveyed survivors of all ages and found the average sample size to be 29.2. Out of 119 articles on minor sex trafficking published between 2000 and 2017 reviewed by Twis and Shelton [15], only 10 peer-reviewed articles were quantitative analyses of datasets representing survivors. Of those 10, all were cross-sectional, and the average sample size was approximately 87. A systematic review by Franchino-Olsen [16] of studies addressing the CSE of children found six quantitative and qualitative peer-reviewed articles with an average sample size of approximately 145. In addition to small sample sizes, the field of inquiry also lacks studies that are rooted in theory [15, 17].

## Theory

As sex trafficking research is still a relatively nascent field of study, testable, theoretical models of CSE do not yet exist. Out of the 10 relevant articles reviewed by Twis and Shelton [15], half of them did not consider theory at all. Though theory is not necessary for practitioners, policy-makers, and other applied professions to find research into CSE useful [18], theory-driven investigations with clear research questions and hypotheses are critical future directions for the field of study to mature and garner the credibility necessary to decrease rates of victimization and provide quality services to survivors [15].

**Multisystem frameworks.** Where theory does exist, it was developed through qualitative observations and is generally focused on the macro, or societal-level, perspective rather than the micro, or individual-level, perspective [15, 17]. Examples of macro theories include feminist theory, intersectionality, and the political economy perspective [19–31]. These theories have previously framed understandings of intimate partner violence. For an extended literature review of theoretical frameworks around commercial sexual exploitation, see S1 File.

*Ecological model.* Human behavior theory that considers agency embedded within context is necessary to fully understand vulnerability. Bronfenbrenner's [32] ecological theory is a theoretical framework that explains the ways in which individual, relational (i.e., familial or microsystem), social (i.e., community or exosystem), and societal (i.e., cultural experiences or macrosystem) factors shape the individual within it. Individuals are impacted by their own personal attributes, like biology, mental health, educational achievement, and past experiences. Push and pull factors related to CSE at the relational level might include childhood sexual abuse, number of children, and low educational achievement. Number of arrests could be a factor that creates vulnerability for sexual exploitation at the community level. Educational achievement may also be considered at the community level if the reason for low educational achievement was low quality education in the individual's community. Though the macro level-factors are not represented in this present study, it is acknowledged that the idea of agency exists within all of these contexts. Age of entry into CSE and length of CSE could be relevant variables at the Chronosystem level.

## Cycling

Push and pull factors related to CSE may be cyclical throughout a person's lifespan. The phenomenon has been studied within the context of intimate partner violence [33]. "Cycling" is a term originally used to research and describe the "on-again, off-again" nature of romantic relationships not necessarily marked by violence [34]. An example of cycling is the commonly cited statistic that it takes seven times for a victim to leave an abusive relationship [35]. This statistic has been appropriated by advocates for victims of CSE as intimate partner violence is the most closely linked and broadly researched phenomenon similar to sexual exploitation. Defining cycling becomes problematic in the case of CSE due to the paradox between agency and victimization [21, 23, 25, 27, 29, 36, 37]. Sex trafficking occurs when there is an absence of

choices. Conversely, the agency of survivors of trafficking should not be diminished. The cycling phenomenon always exists within context. An unpublished manuscript by the first author and Dr. Lauren Rhulmann analyzed a survey of 120 adult survivors of sex trafficking in the United States and found that the total number of times the survivor attempted to exit or successfully exited exploitation ranged from 0 to 25 times, with a mean of 2.58 times.

## Age of entry and exit

The average age of entry into CSE in the United States is often cited to be between the ages of 12 and 14 [8, 38]. These numbers originate from the juvenile justice system and other samples that represent victims who are still minors. For adults, the average age of entry is between the ages of 18 to 20 years old [7, 39]. The Counter Trafficking Data Collaborative dataset, the largest human trafficking open-source dataset representing 156,330 individual cases of global human trafficking, also identifies the average age of entry into exploitation for females as being between the ages of 18 and 20 (CTDC, 2023) [40]. The same dataset suggests that the average length of exploitation is around 2 years. However, Kramer [41]. found that the length of exploitation for American females ranged from 2 months to 42 years with the average length of exploitation of 8.7 years.

## Childhood sexual abuse

The Centers for Disease Control and Prevention (CDC) through its ACES Study estimated that 1 and 4 girls and 1 and 6 boys in the United States are sexually abused before the age of 18 [42]. Survivors of CSE report a rate of childhood sexual abuse between 60% and 95%, though the true average is estimated to be closer to 85% [3]. Several studies have identified a history of childhood sexual abuse as one of the primary risk factors for CSE [3–5, 8]. For survivors of CSE who report a history of childhood sexual abuse, 70% report that it influenced their entry into CSE [8].

## Barriers to exiting

Though 89% of victims of CSE express a desire to exit exploitation, a gap exists between a victim's present reality of exploitation and her ability to access services [3]. The United Nations Office on Drugs and Crime [43] estimates that victims of CSE often attempt to access services, but only 1 in 100 is able to do so. Multiple access barriers are present including documentation or identification, emergency medical and dental care, mental health treatment, drug and alcohol treatment, program costs, criminal histories, and comprehensive program applications and screenings [44, 45]. Victims of CSE in the United States expressed that: 78% needed a safe home or safe place; 73% needed job training; 61% needed healthcare; 56% needed counseling; 67% needed drug/alcohol treatment; and 23% needed physical protection from an exploiter (pimp) [3]. Unfortunately, trafficking victims reported only seeing three ways of exiting exploitation: 1) Become unprofitable (via mental illness or advanced pregnancy); 2) be assisted by a client, or 3) die [12].

**Educational achievement.** The United States Census Bureau [46] reports that 8.9% of the general public have less than a high school diploma or equivalent. In a sample of adult female survivors of prostitution ($N$ = 96), Cronley et al. [2] found that 26.9% had less than a high school diploma. Low educational achievement serves as a barrier toward legitimate, skilled employment options. Therefore, low educational achievement may serve as a push factor for entry into CSE [2, 3, 7, 8]. In addition, restricted access to education may limit knowledge of risks associated with CSE, creating a barrier to services and limiting intervention opportunities.

**Incarceration and number of arrests.** Since ancient times, governments have attempted to regulate (and profit from) sexual exploitation [47, 48]. The Selective Service Act of 1917 was the first movement in the United States to criminalize women for being prostituted because of the spread of venereal disease among U.S. soldiers. Since then, approximately 90,000 prostitution-related arrests are made each year [48, 49]. Research suggests that law enforcement struggle to identify victims and perpetrators of sex trafficking, with an accuracy rate of only 35.7% [50]. A study by Koegler et al. found that a history of prostitution-related arrests predicted further prostitution-related arrests [51]. Along with lost time spent in jail, additional consequences can include crippling fees, lost employment opportunities, a loss of parental rights, and sex offender registration. Although prostitution-related arrests are trending downward since the passage of the Trafficking Victims Protection Act arrest may actually provide an "off-ramp" out of sexual exploitation via separation from a trafficker (i.e., pimp or controller), forced sobriety, or alternative sentencing [1, 52].

**Mental health diagnoses.** Mental health (i.e., mental health disorders, mental illness, etc.) is a broad term used to describe a variety of conditions that affect thought, mood, and behavior. Types of mental health disorders reported included anxiety, bipolar disorder, depression, neurodevelopmental disorders, posttraumatic stress disorder (PTSD), and schizophrenia spectrum disorder. The National Institute of Mental Health estimates that one in five American adults lives with at least one mental health diagnosis [53]. However, disproportionate numbers of survivors of CSE have been diagnosed with one or more mental health conditions even when compared to other high-risk populations, like runaways, juvenile offenders, and children in foster care [54]. Depending on the specific condition, mental health diagnoses may serve as both a predictor (as may be the case for disorders like bipolar disorder and psychotic disorders) for or outcome (as in anxiety, depression, and trauma-related disorders) of CSE [12, 54–56]. Unfortunately, there is limited research that is focused on the relationship between mental health disorders and CSE [57]. This investigation will consider diagnoses of bipolar disorder, neurodevelopmental disorders, and schizophrenia spectrum disorders as predictors in the models. For an extended literature review of mental health diagnoses and commercial sexual exploitation, see S1 File.

**Motherhood and number of children.** Pew Research Center estimates that 86% of women in the United States have had one or more children by the end of their childbearing years [58]. According to the World Bank, the average number of births per female in the United States is 1.6 births [59]. For individuals who have experienced CSE, having children may serve as a protective factor or establish further vulnerability for ongoing exploitation.

Many survivors of CSE cite their desires to be a good mother and role model to their children as a catalyst for exiting exploitation, finding alternative work, and completing drug and alcohol treatment [60]. Unfortunately, many victims feel as if sexual exploitation is the only way for them to support their children, and some even identify their children as the primary push factor for entry into CSE [60, 61]. In a qualitative study of U.S. victims of CSE ($N = 16$), Sloss and Harper found that victims reported increased stress and anxiety after having children [61]. A mother's involvement in CSE has the potential to expose children to violence, abuse, addiction, and exploitation [60, 61]. A Canadian study found that 38% of victims of CSE have lost custody of their children [62]. A follow-up study found that 13% referenced fear of losing custody of their children as a barrier to services [63]. The loss of custody of their children is cited as an additional push factor into exploitation, reducing responsibilities, increasing freedom, and increasing emotional pain that may also lead to increased substance abuse in attempts to self-medicate [61, 64].

## The present study

The present study investigated the secondary dataset of 1,264 unique case files of survivors of CSE collected between 2011 and 2021 by a temporary safe home program located in Atlanta, GA. There are six hypotheses associated with the investigation. It was hypothesized that lower educational achievement, the experience of childhood sexual abuse, mental health diagnoses (specifically bipolar disorder, neurodevelopmental disorders, or schizophrenia spectrum disorder), younger ages of entry into CSE, longer experiences of CSE, higher number of arrests, and higher number of children will predict:

- More instance of cycling,

- Younger age of entry into CSE,

- Longer experiences of CSE,

- Higher number of arrests,

- Higher number of children, and

- More negative placement outcomes.

## Methods

### Procedures

Data were collected between 2011 and 2021 by a temporary safe home program located in Atlanta, GA. Relevant datapoints include educational achievement, prevalence of childhood sexual abuse, mental health diagnoses (bipolar disorder, neurodevelopmental disorders, and schizophrenia spectrum disorders), age of entry into CSE, number of arrests, number of children, and number of times entering the safe home program (i.e., cycling) along with relevant placement notes. Data were collected based on self-reports given to various case managers as part of an intake interview. These data were used with permission from Frontline Response International, Inc. All research activities were conducted under the oversight of Auburn University's Institutional Review Board (IRB) under protocol #22–192. This secondary analysis of existing data was granted a waiver of consent by Auburn University's IRB, and researchers accessed the data on April 27, 2022.

Since 2011, data collection procedures have grown and become more sophisticated as the organization matured and granting bodies increased tracking and reporting requirements. Between 2011 and 2014, virtually no data were collected on survivors entering the safe home program beyond transportation and placement notes. In 2015, case managers began tracking disability, mental health diagnoses, history of childhood sexual abuse, and type of exploitation. By 2018, the organization began receiving funding associated with the Victims of Crimes Act [65], which provided an additional boost to the program's tracking requirements. Because of these changes, there are substantial missing data for certain variables. Additional critical measures have been implemented recently, such as pre/post Likert-style questionnaires and self-report exit surveys. However, these are not considered in the current study because there is not a large enough sample size for analysis at this time.

### Safe home program description

The goal of the anti-trafficking organization was to provide holistic, individualized, and trauma-informed care to adult survivors of CSE through a continuum of services including: Outreach, jail mentorship, 24-hour hotline services, safe home services, and long-term

program placement. The organization operated a crisis safe home for female victims of sex trafficking and CSE. Program participants were at least 18 years old and biologically female, disclosed a history of CSE, did not have any outstanding arrest warrants, were willing to enter a long-term program, and agreed to follow safe home guidelines necessary for the safety of residents and staff. Residents stayed in the safe home for 2–4 weeks while attending medical and psychiatric appointments, participating in group therapy and skills-building classes, and applying for long-term programs. The safe home was located in an undisclosed location and accepted emergency intakes around the clock. Utilizing a team-based approach, professional staff provided trauma-informed care as they conducted and initial intake, established goals, and built an Individualized Service Plan (ISP). With a primary goal of crisis stabilization, all safe home programming, home environment, and staff methods focused on establishing safety and improving emotional self-regulation and posttraumatic stress awareness. Practical goals often included attending court hearings, receiving medical evaluations, and procuring government identification.

The organization partnered with over 40 long-term programs nationwide to offer each resident multiple program options to address her needs. Lengths of long-term programs varied from 90 days to 2 years, though programs that were at least one-year in length were preferred. After acceptance to a long-term program, the organization financially provided for all program fees. From 2011 to 2021, the safe home averaged 164 intakes per year, and over 1,264 unique women have been served in the safe home program.

## Participants

Study participants ($N$ = 1,264) were adult female survivors of CSE who participated in a safe home program in Atlanta, GA between 2011 and 2021. Program participants were referred to the program by law enforcement, Homeland Security, healthcare workers, attorneys, friends and family, and various outreach initiatives. All victims interested in services contacted a hotline for a brief intake to ensure program eligibility.

Program participants were between the ages of 16 and 69, with an average age of 32.58 ($SD$ = 10.59) at time of entry into the safe home program. Although survivors must be 18 to be eligible for services, there are a few cases where 16- and 17-year-olds received services in the safe home. These cases were included in analyses.

Seventy-five percent of the sample met the federal definition for sex trafficking victimization, and 33.4% entered CSE before the age of 18 years old. More than 70% of the sample report a history of childhood sexual abuse with an average age of first experience of sexual abuse of 8 years old. Approximately half (51.2%) of the sample identified as White, 40.4% identified as Black or African American, 2.8% identified as Hispanic (Latina), 0.3% identified as American Indian (Native American), 0.3% identified as Asian, 0.1% identified as Middle Eastern, 3.9% identified as multiracial, and 0.8% identified as Other. About 4.1% of the sample reported their highest level of education as a bachelor's degree or more, 19.6% some college, 16.5% a high school diploma, 15.9% a General Equivalency Degree (GED), 31.2% less than a high school diploma/GED, 6.0% middle school, 0.3% elementary school, and 0.6% no education. The average number of times entering the safe home was 1.28 times ($min$ = 1.00, $max$ = 8.00). Age of entry into CSE ranged from 2 to 55 years old with an average age of 22.07 ($SD$ = 8.67). Total length of time in sexual exploitation ranged from less than one year to 46 years, with the mean of 11.73 years ($SD$ = 10.40). Due to data loss, case managers were only able to retain data from 38% of participants about the location of their first experience of exploitation. Thirty-four out of 50 states were represented, including Florida (18.0%), Tennessee (12.0%), North Carolina (10.9%), South Carolina (6.5%), California (4.9%), Illinois (4.9%),

**Table 1. Sample demographics.**

| | Age of first experience of sexual abuse | Age | Age of entry | Number of arrests | Number of children | Cycling (Number of times in safe home) | Length of exploitation |
|---|---|---|---|---|---|---|---|
| Mean | 7.97 | 32.58 | 22.07 | 9.52 | 1.88 | 1.28 | 11.73 |
| Median | 7 | 31 | 19 | 5 | 2 | 1 | 9 |
| Mode | 5 | 25 | 18 | 3 | 0 | 1 | 2 |
| Min | 0 | 16 | 2 | 0 | 0 | 1 | 0 |
| Max | 17 | 29 | 55 | 100 | 10 | 8 | 46 |
| Standard Deviation | 3.996 | 10.588 | 8.670 | 13.267 | 1.765 | 0.690 | 10.403 |

New York (4.9%), Texas (4.3%), Alabama (3.8%), Pennsylvania (3.3%), Louisiana (2.2%), and others. See Table 1 for sample demographics. See Table 2 for Sample Descriptives and Table 3 for Mean Comparisons with U.S. General Population, Survivors of CSE, and the Present Study.

## Measures

**Childhood sexual abuse.** Survivors were asked if they had ever experienced childhood sexual abuse and, if so, age of childhood sexual abuse. Data were entered into a single field. While most reported a single age, some reported an age range. Therefore, two new variables were created. The first variable captured the experience of childhood sexual abuse as Yes (coded as 1) or No (coded as 0). The second variable represented the age of first experience of childhood sexual abuse. For those who reported an age range, the first age is listed in the new variable. For individuals who listed "infant" as their age of first experience of childhood sexual abuse, it was estimated as being 1 year of age.

**Educational achievement.** Case managers logged educational achievement based on survivor self-report. For descriptives, the investigator grouped responses based on an ordinal scale of "No Education," "Elementary Education," "Middle School Education," "Less than High School Diploma/GED," "GED," "High School Diploma," "Technical School," "Some College," "Associate Degree," "Bachelor's Degree," and "Master's Degree" and coded them 0–11, respectively.

**Cycling.** Cycling was determined by the total number of times the survivor entered the safe home.

**Age of entry and exit.** Age of entry is the age of first exploitation which was a self-reported age of the first time the survivor was given something of value in exchange for a sex act. Age of exit is determined by the reported age of the survivor at time of final entry into the safe home.

**Length of exploitation.** Length of exploitation was determined by creating a new variable and subtracting age of entry into sexual exploitation from age of final entry into the safe home.

**Number of arrests.** Incarceration was operationalized by the total number of arrests, an estimated self-report number that was logged by a case manager.

**Mental health diagnoses.** Survivors were asked if they had ever been diagnosed with a mental health condition, coded as Yes (coded as 1) or No (coded as 0). Then, survivors were asked to list their mental health diagnoses which were all entered into the same field as qualitative data. The investigators created new variables representing each mental health diagnosis, including anxiety, bipolar disorder, depression, neurodevelopmental disorders, posttraumatic stress disorder, and schizophrenia spectrum disorders, and entered whether the survivor had (coded as 1) or had not (coded as 0) reported that they had been diagnosed with each mental

**Table 2. Descriptive statistics.**

| | Ever Experience Childhood Sexual Abuse | Age of Entry | Anxiety | Arrests | Bipolar Disorder | Number of Children | Cycling | Depression | Educational Achievement | Length of Exploitation | Neurodevelopmental Disorder | Placement | PTSD | Schizophrenia Spectrum Disorder | Trafficking |
|---|---|---|---|---|---|---|---|---|---|---|---|---|---|---|---|
| | (n = 723) | (n = 712) | (n = 794) | (n = 633) | (n = 792) | (n = 729) | (n = 1264) | (n = 797) | (n = 734) | (n = 704) | (n = 797) | (n = 1245) | (n = 797) | (n = 797) | (n = 720) |
| Mean | .710 | 22.070 | .330 | 9.520 | .380 | 1.880 | 1.277 | .420 | 4.640 | 11.730 | .110 | .550 | .340 | .130 | .750 |
| Std. Error of Mean | .017 | .325 | .017 | .527 | .017 | .065 | .019 | .017 | .730 | .392 | .011 | .014 | .017 | .012 | .016 |
| Median | 1.000 | 19.000 | .000 | 5.000 | .000 | 2.000 | 1.000 | .000 | 4.000 | 9.000 | .000 | 1.000 | .000 | .000 | 1.000 |
| Mode | 1.000 | 18.000 | .000 | 3.000 | .000 | .000 | 1.000 | .000 | 3.000 | 2.000 | .000 | 1.000 | .000 | .000 | 1.000 |
| Std. Deviation | .456 | 8.670 | .469 | 13.267 | .486 | 1.765 | .690 | .494 | 1.975 | 10.403 | .307 | .498 | .474 | .334 | .433 |
| Variance | .208 | 75.162 | .220 | 176.022 | .236 | 3.115 | .476 | .244 | 3.901 | 108.229 | .094 | .248 | .224 | .112 | .188 |
| Skewness | -.910 | .992 | .743 | 3.657 | .490 | 1.109 | 3.351 | .323 | .476 | .931 | 2.575 | -.189 | .683 | 2.231 | -1.157 |
| Std. Error of Skewness | .091 | .092 | .087 | .097 | .087 | .091 | .069 | .087 | .090 | .092 | .087 | .069 | .087 | .087 | .091 |
| Range | 1.000 | 53.000 | 1.000 | 100.000 | 1.000 | 10.000 | 1.000 | 1.000 | 10.000 | 46.000 | 1.000 | 1.000 | 1.000 | 1.000 | 1.000 |
| Minimum | .000 | 2.000 | .000 | .000 | .000 | .000 | 1.000 | .000 | .000 | .000 | .000 | .000 | .000 | .000 | .000 |
| Maximum | 1.000 | 55.000 | 1.000 | 100.000 | 1.000 | 10.000 | 8.000 | 1.000 | 10.000 | 46.000 | 1.000 | 1.000 | 1.000 | 1.000 | 1.000 |

**Table 3. Mean comparisons with U.S. general population, survivors of Commercial Sexual Exploitation (CSE), and the present study.**

| | General U.S. population | Literature related to survivors of CSE | The present study |
|---|---|---|---|
| Ever experience childhood sexual abuse | .250 | .850 | .710 |
| Age of entry | - | 18–20 | 22.070 |
| Anxiety | .311 | .009 - .196 | .330 |
| Bipolar disorder | .044 | .266 | .380 |
| Number of children | 1.600 | - | 1.880 |
| Cycling | - | 2.580 | 1.277 |
| Depression | .084 | .273 - .455 | .420 |
| Educational Achievement | | | |
| Less than high school diploma or equivalent | .089 | .269 | .381 |
| High school diploma or equivalent | .279 | .495 | .324 |
| Some college | .149 | .237 | .196 |
| Associate degree | .105 | | |
| Bachelor's degree | .235 | | .041 |
| Advanced degree | .144 | | |
| Length of exploitation | - | 8.700 | 11.730 |
| Neurodevelopmental disorders | - | - | .110 |
| ADHD | .056 | .524 | .084 |
| Autism spectrum disorder | .009 | - | .009 |
| Dyslexia | .03 - .07 | - | .013 |
| Unspecified learning or cognitive disability | - | - | .015 |
| PTSD | .068 | .890 | .340 |
| Schizophrenia spectrum disorder | .003 - .006 | .140 | .130 |
| Trafficking | - | .840 | .750 |

*Citations for this table are available in S1 Table.

health disorder. Neurodevelopmental disorders was a combination of ADHD, Autism Spectrum Disorder, dyslexia, and other unspecified learning or cognitive disability diagnoses to represent potential barriers in educational and occupational achievement. Bipolar disorder, neurodevelopmental disorders, and schizophrenia spectrum disorders were used as predictors in regression analyses.

**Number of children.** Survivors were asked, "Are you a mother? If so, how many children do you have?" Answers were self-reported and logged into a single field by a case manager. Two new variables were created. The first variable represented whether the survivor was a mother, as Yes (coded as 1) or No (coded as 0). The second variable was a count of the total number of children.

**Placement.** Placement into a long-term program is a primary goal for the safe home program. All survivors entering the program must agree to long-term program placement when they call the hotline in order to be accepted into the safe home. Placement options included long-term placement (coded as 1) or no long-term placement (coded as 0). No long-term placement can represent survivors who were hospitalized for mental health or substance use disorders, those who needed to address outstanding warrants, individuals who changed their minds about long-term program placement, or those who left prematurely for other various reasons.

## Plan of analysis

**Preliminary analyses.** First, descriptive statistics and frequencies of predictors and outcomes were evaluated using the Statistical Package for Social Sciences (SPSS) version 26 [66]. Missing data were coded as -99 and addressed using Full Information Maximum Likelihood estimation (FIML) in regression analyses as appropriate [67]. Given the large amount of missing data, the investigator evaluated the data for patterns of missingness by creating a new variable representing the total number of missing variables for each case. Then, regression analyses were conducted to evaluate whether any of the dependent variables were related to the total amount of missing variables. No significant associations were found.

**Primary analyses.** Regression analyses were conducted using MPlus to evaluate associations between predictors and outcomes [68]. Based on preliminary analyses of the means, variances, and distributions of the outcome variables, the most appropriate type of regression analysis was chosen. Linear regression was selected in cases where the outcome was normally distributed. Zero-inflated Poisson regression was selected where the outcome is right skewed with many zeros. Binomial logistic regression was applied to nominal outcomes with two categories (i.e., yes, no, for placement).

## Results

### Preliminary analyses

Bivariate correlations examined the associations between the study variables (Table 4). Cycling was significantly positively correlated with number of arrests ($r = .092$, $p = .021$), bipolar disorder ($r = .160$, $p < .001$), number of children ($r = 0.084$, $p = .023$), and length of CSE ($r = .114$, $p = .003$) and negatively correlated with age of entry into CSE ($r = -.088$, $p = .019$). Age of entry into CSE was significantly positively correlated with educational achievement ($r = .242$, $p < .001$) and number of children ($r = .090$, $p = .018$) and negatively correlated with childhood sexual abuse ($r = -.166$, $p < .001$), length of CSE ($r = -.425$, $p < .001$), and schizophrenia spectrum

**Table 4. Bivariate correlation table.**

| | Age of Entry | Number of Arrests | Bipolar Disorder | Number of Children | Cycling | Educational Achievement | Length of Exploitation | Neurodevelopmental Disorder | Placement | Schizophrenia Spectrum Disorder |
|---|---|---|---|---|---|---|---|---|---|---|
| Ever Experience Childhood Sexual Abuse | -.166*** | .007 | .055 | .027 | .070 | -.090* | .082* | .027 | .026 | .054 |
| Age of Entry | - | -.048 | -.050 | .090* | -.088* | .242*** | -.425*** | -.052 | .070 | -.087* |
| Arrests | | - | -.004 | .171*** | .092* | -.092 | .296*** | -.053 | .052 | .023 |
| Bipolar Disorder | | | - | .022 | .160*** | -.061 | .067 | .118*** | .014 | .187*** |
| Number of Children | | | | - | .084* | -.089* | .303*** | -.085* | -.021 | .076* |
| Cycling | | | | | - | -.034 | .114** | .054 | .018 | .036 |
| Educational Achievement | | | | | | - | -.043 | -.019 | .042 | -.048 |
| Length of Exploitation | | | | | | | - | -.017 | -.023 | .120*** |
| Neurodevelopmental Disorder | | | | | | | | - | -.082* | .089* |
| Placement | | | | | | | | | - | -.122*** |

*Correlation is significant at the .05 level (2-tailed).

**Correlation is significant at the .01 level (2-tailed).

***Correlation is significant at the .001 level (2-tailed).

disorder ($r$ = -.087, $p$ = .021). Length of CSE was significantly positively correlated with childhood sexual abuse ($r$ = .082, $p$ = .032), number of arrests ($r$ = .296, $p < $ .001), number of children ($r$ = 0.303, $p < $ .001), and schizophrenia spectrum disorder ($r$ = .120, $p < $ .001). Number of children was significantly positively correlated with number of arrests ($r$ = .171, $p < $ .001) and schizophrenia spectrum disorder ($r$ = .076, $p$ = .042) and negatively correlated with educational achievement ($r$ = -.089, $p$ = .018) and neurodevelopmental disorders ($r$ = -.085, $p$ = .022). Lastly, the variable, placement outcomes, was significantly negatively correlated with neurodevelopmental disorders ($r$ = -.082, $p$ = .021) and schizophrenia spectrum disorder ($r$ = -.122, $p$ = .001).

## Primary analyses

All models were saturated. Therefore, fit indices are not reported for any of the models. See Tables 5 through 7 for Model Results.

**Cycling.** The median number of times an individual entered the safe home was 1, and the data were positively skewed with a visible tail in the positive direction ($min$ = 1.000; $max$ = 8.000; $skew$ = 3.351; $se$ = .069). Approximately 18.6% of the sample experienced at least one episode of cycling. Because of this non-normality, a new variable was created where the total number of times in the safe home was subtracted by one to make it suitable for zero-inflated Poisson regression analysis. Educational achievement, childhood sexual abuse, mental health diagnoses, age of entry into CSE, length of CSE, number of arrests, and number of children were used as predictors of cycling. A diagnosis of bipolar disorder ($B$ = .887, $p < $ .001) and neurodevelopmental disorders ($B$ = .556, $p$ = .040) were positively related to the log count of cycling (Table 5). Individuals who self-report a diagnosis of bipolar disorder were 2.51 times more likely to experience cycling, and individuals with a self-reported neurodevelopmental disorder were 1.73 times more likely to experience cycling. Further, individuals with higher educational achievement and higher number of arrests were less likely to experience cycling.

**Age of entry.** The mean age of entry into CSE was 22.07 years old ($SD$ = 8.670; $SEM$ = .325). Age of entry into CSE was normally distributed, and a linear regression model was fit to determine the relationship between educational achievement, experience of childhood sexual

**Table 5. Zero-inflated Poisson regression model results for outcome variable, cycling.**

|  | Cycling | Cycling # |
|---|---|---|
| Ever Experience Childhood Sexual Abuse | .129 | -.160 |
| Age of Entry | .004 | .039 |
| Arrests | -.003 | -.030* |
| Bipolar Disorder | .920*** | .516 |
| Number of Children | .007 | -.076 |
| Educational Achievement | -.110 | -.191* |
| Length of Exploitation | .015 | .016 |
| Neurodevelopmental Disorder | .547* | .352 |
| Schizophrenia Spectrum Disorder | -.508 | -.762 |
| $R$-squared | - | .116[a] |

Results reported are unstandardized.

*Indicates significance at the .05 level.

**Indicates significance at the .01 level.

***Indicates significance at the .001 level

[a]p-value not reported.

**Table 6. Linear regression model results for outcome variables, age of entry, length of exploitation, number of arrests, and number of children.**

| | Age of Entry | Length of Exploitation | Number of Arrests | Number of Children |
|---|---|---|---|---|
| Ever Experience Childhood Sexual Abuse | -2.530*** | -.016 | -.247 | .120 |
| Age of Entry | - | -.560*** | .198** | .062*** |
| Arrests | - | .182*** | - | - |
| Bipolar Disorder | -.192 | .747 | -.520 | -.007 |
| Number of Children | - | 1.796*** | - | - |
| Educational Achievement | .993*** | .673*** | -.818** | -.128*** |
| Length of Exploitation | - | - | .443*** | .070*** |
| Neurodevelopmental Disorder | -.989 | -.282 | -1.853 | -.383* |
| Schizophrenia Spectrum Disorder | -1.586 | 1.657 | -.150 | .269 |
| *R*-squared | .083*** | .362*** | .112*** | .173*** |

Results reported are unstandardized.

*Indicates significance at the .05 level.

**Indicates significance at the .01 level.

***Indicates significance at the .001 level

abuse, mental health diagnoses, and the outcome variable, age of entry. The model results suggested that the experience of childhood sexual abuse ($B$ = -2.530, $p < .001$) predicted younger ages of entry into CSE, and higher educational achievement ($B$ = .993, $p < .001$) predicted older ages of entry into CSE ($R^2$ = .083, $p < .001$; Table 6).

**Length of exploitation.** Length of exploitation had a mean value of 11.73 ($SD$ = 10.403; $SEM$ = .392) years and was normally distributed. A linear regression analysis used educational achievement, childhood sexual abuse, mental health diagnoses, age of entry into CSE, number of arrests, and number of children as predictors of length of CSE. Results indicate lower educational achievement ($B$ = .673, $p < .001$), younger entry into CSE ($B$ = -.560, $p < .001$), higher number of arrests ($B$ = .182, $p < .001$), and higher number of children ($B$ = 1.796, $p < .001$) predicted longer experiences of CSE ($R^2$ = .362, $p < .001$; Table 6).

**Number of arrests.** The majority of the sample (90.7%) had been arrested at least one time, and the mean number of arrests was 9.52 ($SD$ = 13.267; $SEM$ = .527). A normal distribution was determined, and a linear regression model used educational achievement, childhood sexual abuse, mental health diagnoses, age of entry into CSE, and length of CSE as predictors of number of arrests. Lower educational achievement ($B$ = -.818, $p = .002$), younger ages of entry ($B$ = .198, $p = .001$), and longer experiences of CSE ($B$ = .443, $p < .001$) predicted higher number of arrests ($R^2$ = .112, $p < .001$; Table 6).

**Number of children.** A large portion of the sample (72.3%) reported being a mother. The median number of children birthed was 2 ($min$ = 0.000; $max$ = 10.000; $skew$ = 1.109; $se$ = .091), and the data were slightly positively skewed with a visible tail in the positive direction, though not skewed enough to warrant the use of Poisson regression. Thus, educational achievement, childhood sexual abuse, mental health diagnoses, age of entry into CSE, and length of CSE were used as predictors of number of children in the linear regression model. Results indicate older ages of entry into CSE ($B$ = .062, $p < .001$), lower educational achievement ($B$ = -.128, $p < .001$), and longer experiences of CSE ($B$ = .070, $p < .001$) predicted a higher number of children, and a diagnosis of a neurodevelopmental disorder ($B$ = -.383, $p = .048$) predicted a lower number of children birthed ($R^2$ = .173, $p < .001$; Table 6).

**Table 7. Binomial logistic regression model results for outcome variable, placement.**

|  | Placement |
| --- | --- |
| Threshold | .274 |
| Ever Experience Childhood Sexual Abuse | .180 |
| Age of Entry | .013 |
| Arrests | .008 |
| Bipolar Disorder | .108 |
| Educational Achievement | .013 |
| Length of Exploitation | .001 |
| Neurodevelopmental Disorder | -.693* |
| Schizophrenia Spectrum Disorder | -.713** |
| R-squared | .045* |

Results reported are unstandardized logistic regression coefficients.

*Indicates significance at the .05 level.

**Indicates significance at the .01 level.

***Indicates significance at the .001 level

**Placement.**   Finally, a binomial logistic regression model using educational achievement, childhood sexual abuse, mental health diagnoses, age of entry into CSE, length of CSE, and number of arrests as predictors of placement. Results indicate that individuals with a neurodevelopmental disorder ($B$ = -.693, $p$ = .016) or schizophrenia spectrum disorder ($B$ = -.713, $p$ = .005) were about half as likely to be placed into a long-term program compared to those who did not report a neurodevelopmental disorder or schizophrenia spectrum disorder (odds ratios = .50 and .49, respectively; $R^2$ = .045, $p$ = .023; Table 7).

## Discussion

The study utilized one of the largest known datasets representing survivors of CSE, allowing for empirical findings that will inform policy, strengthen practice, and address a significant dearth in the literature. Though cross-sectional in nature, the relatively large dataset allows for greater generalizability of the findings. The investigation considered associations between relevant predictor variables and exploitation measures including cycling, age of entry into CSE, length of CSE, number of arrests, number of children, and placement outcomes. The findings provide a portrait of exploitation across a lifespan that was previously unavailable with smaller datasets. Preliminary results suggest an older average age of entry (22.07 years old) and longer experience of exploitation (11.73 years) than previously reported in the literature. Additionally, the prevalence of sex trafficking victimization (75%) suggests that exploited individuals are more likely victims of crime than criminals [52]. Preliminary findings go on to highlight significant mental health considerations within this population compared to the general population (80% rate mental health diagnoses compared to a 20% rate) [53].

### Barriers to getting out and staying out

The results of this study highlight just a few of the legitimate barriers for survivors of CSE—both barriers to getting out and barriers to staying out. A key takeaway from the results is that educational achievement may serve as an important protective factor as it was linked to older ages of entry into CSE, shorter lengths of CSE, lower arrest rates, and lower number of children. Low educational achievement limits access to employment and economic opportunities. As the sample consisted of individuals who have all experienced exploitation, it cannot be

stated that education can prevent exploitation. However, findings indicate that education may decrease the severity of exploitation if severity is defined as earlier ages of entry and longer experiences of exploitation.

An additional barrier to getting out and staying out of exploitation featured in this study is a history of arrests. A criminal history can put a profound burden on an individual's resources and act as a significant barrier to services, employment, welfare, and housing. Results of this investigation indicate that more arrests are linked to longer experiences of exploitation. Alternatives to arrest, like alternative sentencing or connecting a victim directly to services, may be more beneficial for survivors. There are also legislative opportunities to address this particular barrier, including expungement and vacatur. Vacatur allows survivors to have their convictions vacated if their crimes were committed while they were being trafficked. Similarly, expungement permanently seals or destroys one's arrest record if they are identified as a victim of sex trafficking.

The literature suggests that children may be a primary push factor for CSE and fear of losing custody of their children is a barrier to services [60, 61, 63]. The findings in this investigation support this wherein number of children is positively related to longer experiences of exploitation. Individuals with children may require unique considerations to decrease severity of exploitation, including affordable childcare and transportation and programs allowing children to reside with their mothers while in care. Further, universal healthcare and improved access to contraceptives may decrease severity of exploitation.

Lastly, the findings of this study call attention to vulnerabilities related to mental health disorders. Individuals with a history of CSE have a rate of mental health disorders that is four times the national rate [53]. Results further suggest that diagnoses of bipolar disorder and neurodevelopmental disorders were related to higher rates of cycling, and neurodevelopmental disorders and schizophrenia spectrum disorders were related to poorer placement outcomes. Practitioners need to be prepared to serve such individuals to improve placement outcomes and disrupt the cycling phenomenon. Training for staff and allowance of the use of psychiatric medications while in programs are critical steps in supporting these survivors. Whenever possible, partnerships with psychiatric services may be a critical component for success. Individuals with a diagnosis of a schizophrenia spectrum disorder or neurodevelopmental disorder (e.g., intellectual disabilities) specifically are uniquely vulnerable to abuse, exploitation, and reexploitation, especially in cases where they do not have a built-in support system, like family [69]. Unfortunately, many long-term programs feel ill-equipped to serve such individuals, making it very difficult for case managers to determine adequate care. While this does beg a conversation about long-term program requirements, it also calls attention to the necessity for more robust, long-term community-based mental health services.

## Theoretical connections

The evidence of such extreme disparities within the population represented in this study in terms of educational achievement, mental health disorders, and abuse histories support a neoabolitionist perspective in that the unequal balance of power negates a woman's ability to truly consent to commercial sex [21, 23, 27, 29]. Results highlight some of the push and pull factors embedded within a multisystem framework that may affect a survivor's ability to exit exploitation [32]. More information related to these factors, as well as others, would be helpful in identifying how these variables relate to one another and the various systems in which individuals are embedded. For example, a better understanding of the level of influence of educational achievement would assist in identifying preventative measures (i.e., individual attributes vs. issues within school systems). Further, educational achievement may not actually be a

protective factor but serve as a proxy for other protective factors, like housing stability, parental support, etc. Future research should isolate education from such factors to determine its relationship to relevant outcome variables. Future research may also consider conditional (i.e., moderated) relationships between the push and pull factors at the various levels. For example, the combination of higher number of children and lower educational achievement could be especially limiting. Another critical future direction for research related to CSE is the development of productive, process-oriented, testable theory aimed toward the conditions that lead people into and out of exploitation [17, 70].

## Limitations

Though the results of this investigation provide novel insight into the experience of CSE, there are several limitations related to the study. First, the investigation relied on self-reported data. As with self-report data, it is difficult to confirm the veracity of the statements. Second, the use of a secondary dataset meant that the investigator could not shape the questions that may have provided more insight into the experiences of survivors of CSE. Third, cross-sectional data cannot assess causation. Fourth, the measure for cycling was limited to the number of entries into this particular organization's safe home program and may not represent the true number of times survivors attempt to exit exploitation. Finally, the sample only represented survivors of CSE who successfully accessed services, and the findings may not represent a significant portion of the population who do not seek, or could not access, assistance.

## Future directions

Results of this investigation address a significant dearth in the literature and present several opportunities for future research. Along with future directions already discussed in the theoretical connections section, longitudinal studies would be instrumental in assessing how processes unfold in entry and exit of CSE and the directions of effects among the key variables. Future research should include diagnostic measures along with self-report data to better understand the prevalence of mental health disorders within this population. In addition, future research may seek to identify evidence-based therapeutic treatments for survivors of CSE to improve outcomes. Finally, shame associated with the cycling phenomenon may be an additional barrier to re-accessing services. Therefore, a more accurate estimate of the prevalence of cycling will assist in destigmatizing the cycling phenomenon and improve service provision to ultimately disrupt cycling incidences.

## Conclusion

The purpose of this investigation was to consider important predictors of outcomes related to CSE. Results assist in developing a narrative about barriers to getting out and staying out of exploitation. Educational achievement was identified as a potential protective factor, while higher number of arrests and higher number of children were related to increased severity of exploitation when severity is defined as earlier ages of entry and longer experiences of CSE. Further, certain mental health disorders were linked to poorer placement outcomes and higher rates of cycling. Implications of the findings include increased services for individuals with children and alternatives to arrest. Practitioners should partner with psychiatric services to improve outcomes for individuals with mental health disorders as well as partner with researchers to analyze existing data and improve data collection procedures. Future research should utilize diagnostic measures along with self-report data and longitudinal data where possible. With these advancements, a more authentic portrait of CSE can be examined, informing services, interventions, and policy and supporting survivors in their promising futures.

## Supporting information

**S1 File. Supplemental literature review.**
(DOCX)

**S1 Table. Mean comparisons with U.S. general population, survivors of Commercial Sexual Exploitation (CSE), and the present study (Table 3) with citations.**
(DOCX)

## Author Contributions

**Conceptualization:** Courtney Furlong.

**Data curation:** Courtney Furlong.

**Formal analysis:** Courtney Furlong.

**Methodology:** Courtney Furlong, Ben Hinnant.

**Supervision:** Ben Hinnant.

**Writing – original draft:** Courtney Furlong.

**Writing – review & editing:** Ben Hinnant.

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
