## [Decision Letter · Decision Letter 0]

4 Jul 2024

PONE-D-24-06079Sex Trafficking Vulnerabilities in Context: An analysis of 1,264 case files of adult survivors of commercial sexual exploitationPLOS ONE

Dear Dr. Furlong,

Thank you for submitting your manuscript to PLOS ONE. After careful consideration, we feel that it has merit but does not fully meet PLOS ONE’s publication criteria as it currently stands. Therefore, we invite you to submit a revised version of the manuscript that addresses the points raised during the review process.

We look forward to receiving your revised manuscript.

Kind regards,

Shivanand Kattimani

Academic Editor

PLOS ONE

Journal Requirements:

2. Please note that your Data Availability Statement is currently missing direct link to access each database. If your manuscript is accepted for publication, you will be asked to provide these details on a very short timeline. We therefore suggest that you provide this information now, though we will not hold up the peer review process if you are unable.

Additional Editor Comments:

Kindly see the Comments from the reviewer;

Along with that my comments can be seen below:

Sex Trafficking Vulnerabilities in Context: An analysis of 1,264 case files of adult survivors of commercial sexual exploitation

study aimed to test whether predetermined set of variables selected from the review of literature can predict five outcomes in a group of survivors of commercial sexual exploitation.

It is based on records.

I agree with investigators that such studies are rare.

Comment 01: Lived experience cannot be stated..as these are from the data form record....where emotions and behaviors, positive and negative attitudes cannot be captured in these data.

Comment 02:Literature review can be cut-dwon: Bring existing knwoledge and the gap that need to be explored in the current study.

Comment 03:Hypothesis can be can be combined into single one: .

We hypothesised that Lower educational achievement, the experience of childhood sexual abuse, mental health diagnoses (specifically bipolar disorder, neuro-developmental disorder, or schizophrenia), younger ages of entry, longer experiences of CSE, higher number of arrests, and higher number of children will predict:

1) more instance of cycling,

2) younger age of entry into CSE,

3) longer experiences of CSE,

4) higher number of arrests,

5) higher number of children,

6)more negative placement outcomes

Comment 04: Under measurement of sexual abuse: For individuals who listed "infant" as their age of first experience of childhood sexual abuse, it was estimated as being 1 year of age..

There is severe problem of recall bias..Minimum age to recall events that are very strange or important is 3 years and adults can clearly recall signifcant events that have happened only after age 7 years .

Comment 05:There is problem with self reported diagnois- as having schizophrenia, or neurodevelopmental disorder (unless medical records shows documentation by Physician or specialist assessment).This is a major limitation of the study.

Comment 06:In primary analysis: Please mention whether assumptions for regression analysis were fulfilled. Please mention on fitness of the model for regression analysis.

Comment 07:Mention of collinearity between the continuous variables and how it was managed.

Comment 08:Any statistical correction of multiple tests done was applied?

Comment 09

In Table 3. Mean Comparisons with U.S. General Population, Survivors of Commercial Sexual Exploitation (CSE), and the Present Study

what are these superscripts?. We dont insert citations for number in the Table.

haedings of the columns: In the survivors of CSE- you can add 'literature related to survivors of CSE', as the last column heading- this present study is also about survivors of CSE.

What are these values or their units?

Reviewers' comments:

Reviewer's Responses to Questions

**Comments to the Author**

1. Is the manuscript technically sound, and do the data support the conclusions?

Reviewer #1: Yes

2. Has the statistical analysis been performed appropriately and rigorously? 

Reviewer #1: Yes

3. Have the authors made all data underlying the findings in their manuscript fully available?

Reviewer #1: No

4. Is the manuscript presented in an intelligible fashion and written in standard English?

Reviewer #1: Yes

5. Review Comments to the Author

Reviewer #1: The current study aimed to develop a deeper understanding of the lived experiences of adult survivors of commercial sexual exploitation. While the topic is interesting for avoiding future commercial sexual exploitation, there remain issues to improve the work.

1. The total length of the manuscript should be constrained to a readable length. Please shorten the review section regarding the literature theories, if no innovative perspectives were provided by the authors.

2. Or, If deemed appropriate, the lengthy literature review may be placed in the supplementary section, leaving only the most important pieces in the main manuscript.

3. I would also suggest adding some flowcharts to present the main theoretical questions concerning the CSE and explain how the methods used by the authors would help to solve the questions.

4. Table 1 needs to be reorganized.

5. Most of the results revealed by the different regression approaches have r2 lower than 50%, the coefficients should be explained with discretion. If similar to the mainstream of the literature research, authors may need to explain this in their limitation sections.

6. The odds should be reported as odds ratios. I don’t think Table 7’s results aligned with the main manuscript, for the odds of neurodevelopmental disorder and schizophrenia spectrum disorder. Please also check the other results. It would be better if authors could place the tables directly under the text where cited.

7. How will the findings from this study help to improve the social environmental and legislation procedures as well as global cooperation to avoid the CSE? I think the authors should make their implications much clearer in the discussion section.

6. PLOS authors have the option to publish the peer review history of their article (what does this mean?). If published, this will include your full peer review and any attached files.

Reviewer #1: No

---

## [Author Response · Author response to Decision Letter 0]

5 Sep 2024

September 5, 2024

Dear Dr. Emily Chenette and the editorial staff at PLOS One,

Thank you to you and the Reviewers for your thoughtful feedback on this manuscript, titled “Sex Trafficking Vulnerabilities in Context: An analysis of 1,264 case files of adult survivors of commercial sexual exploitation.” We have reworked the manuscript in response to the Reviewer's suggestions and find that it is ultimately a better project as a result. The Authors responses to the Reviewers’ specific comments (in italics) are outlined below.

• Reviewer 1 Comment 1: "Lived experience cannot be stated. As these are from the data form record....where emotions and behaviors, positive and negative attitudes cannot be captured in these data." The word, "lived," has been deleted from any instances where "lived experiences" was used.

• Reviewer 1 Comment 2: "Literature review can be cut down. Bring existing knowledge and the gap that needs to be explored in the current study." To address this comment, along with the first comment made by Reviewer 2, the sections discussing theory and mental health diagnoses in the Literature Review were moved to a supplementary section (S1) pending approval by the Editor(s). This edit trimmed the manuscript down approximately 4 pages. 

• Reviewer 1 Comment 3: "Hypothesis can be combined into single one: We hypothesized that Lower educational achievement, the experience of childhood sexual abuse, mental health diagnoses (specifically bipolar disorder, neuro-developmental disorder, or schizophrenia), younger ages of entry, longer experiences of CSE, higher number of arrests, and higher number of children will predict:

o more instance of cycling,

o younger age of entry into CSE,

o longer experiences of CSE,

o higher number of arrests,

o higher number of children,

o more negative placement outcomes" Hypotheses were combined as suggested.

• Reviewer 1 Comment 4: "Under measurement of sexual abuse: For individuals who listed "infant" as their age of first experience of childhood sexual abuse, it was estimated as being 1 year of age. There is severe problem of recall bias. Minimum age to recall events that are very strange or important is 3 years and adults can clearly recall significant events that have happened only after age 7 years." We believe that it is possible that one can have knowledge of early abuse. For example, a friend or family member could remember the abuse, or there could be medical reports or reports made to social services. There is always a limitation when studies rely on self-reported data, and this limitation is noted in the Discussion section of this manuscript. That said, it is critical that we honor the voices and stories of survivors of commercial sexual exploitation and use the data as they are reported.

• Reviewer 1 Comment 5: "There is problem with self-reported diagnosis, as having schizophrenia or neurodevelopmental disorder (unless medical records show documentation by Physician or specialist assessment). This is a major limitation of the study." This limitation is noted in the Discussion section of the manuscript. As stated, there is always a limitation when studies rely on self-reported data. Even so, we endeavor to honor the voices and stories of survivors of commercial sexual exploitation and use the data as they are reported.

• Reviewer 1 Comment 6: "In primary analysis: Please mention whether assumptions for regression analysis were fulfilled. Please mention on fitness of the model for regression analysis." In the Methods section, under the header, "Primary Analyses," assumptions are detailed as follows: "Based on preliminary analyses of the means, variances, and distributions of the outcome variables, the most appropriate type of regression analysis was chosen. Linear regression was selected in cases where the outcome was normally distributed (Table 5). Zero-inflated Poisson regression was selected where the outcome is right skewed with many zeros (Table 6). Binomial logistic regression was applied to nominal outcomes with two categories (i.e., yes, no, for placement; Table 7)." Further, in the Results section under the header, "Primary Analyses," we state that all models were saturated — meaning there were zero degrees of freedom for all models — and that model fit indices cannot be interpreted.

• Reviewer 1 Comment 7: "Mention of collinearity between the continuous variables and how it was managed." In Table 4, binomial correlations are detailed. While there are a few strong correlations, particularly between mental health diagnoses and age and length of exploitation, none of the correlations fall above the 0.8 threshold (Morrissey & Ruxton, 2018; Simon Fraser University, 2011). Therefore, colinearity among predictors was not sufficiently high to be of concern.

o Morrissey, M. B., & Ruxton, G. D. (2018). Multiple regression is not multiple regressions: the meaning of multiple regression and the non-problem of collinearity. Philosophy, Theory, and Practice in Biology, 10(3). https://philpapers.org/archive/MORMRI-4.pdf

o Simon Fraser University (2011). Chapter 8: Multicollinearity [Lecture]. https://www.sfu.ca/~dsignori/buec333/lecture%2016.pdf

• Reviewer 1 Comment 8: "Any statistical correction of multiple tests done was applied?" Type 1 errors are known to be inflated with multiple statistical tests. However, there are known disadvantages to correction of multiple as well, in particular when statistical tests are based on evidence or theory-based hypotheses (Barnett et al., 2022). An over reliance on p-values (and adjusted p-values) can lead to type 2 errors. Thus, a balance between emphasizing traditional null hypothesis significance testing and effect sizes is a healthy middle ground (Dunkler et al., 2020). While we did not adjust p-values for multiple tests, our tests were based on hypotheses derived from logic, prior evidence, and theory - as compared to an exploratory, "fishing" approach (Gelman et al., 2012). Additionally, we report effect sizes so that readers can judge for themselves the meaningfulness of the findings. 

o Barnett, M. J., Doroudgar, S., Khosraviani, V., & Ip, E. J. (2022). Multiple comparisons: To compare or not to compare, that is the question. Research in Social and Administrative Pharmacy, 18(2), 2331-2334. https://doi.org/10.1016/j.sapharm.2021.07.006

o Dunkler, D., Haller, M., Oberbauer, R., & Heinze, G. (2020). To test or to estimate? P‐values versus effect sizes. Transplant International, 33(1), 50-55. https://doi.org/10.1111/tri.13535

o Gelman, A., Hill, J., & Yajima, M. (2012). Why we (usually) don't have to worry about multiple comparisons. Journal of Research on Educational Effectiveness, 5(2), 189-211. https://doi.org/10.1080/19345747.2011.618213

• Reviewer 1 Comment 9: "In Table 3. Mean Comparisons with U.S. General Population, Survivors of Commercial Sexual Exploitation (CSE), and the Present Study. What are these superscripts? We don't insert citations for number in the Table. headings of the columns: In the survivors of CSE- you can add 'literature related to survivors of CSE', as the last column heading- this present study is also about survivors of CSE. What are these values or their units?" Superscripts have been removed. Header was edited to read, "Literature related to survivors of CSE." An asterisk was added to the table description that states, "Citations for this table are available in the References section and S1."

• Reviewer 2 Comment 1: "The total length of the manuscript should be constrained to a readable length. Please shorten the review section regarding the literature theories, if no innovative perspectives were provided by the authors. Or, If deemed appropriate, the lengthy literature review may be placed in the supplementary section, leaving only the most important pieces in the main manuscript." To address this comment, along with the second comment made by Reviewer 1, the sections discussing theory and mental health diagnoses in the Literature Review were moved to a supplementary section (S1) pending approval by the Editor(s). This edit trimmed the manuscript down approximately 4 pages.

• Reviewer 2 Comment 2: "I would also suggest adding some flowcharts to present the main theoretical questions concerning the CSE and explain how the methods used by the authors would help to solve the questions." Since the theory discussion has been moved to the supplemental section per the Reviewer's suggestion, the theories presented in the main text of the document are more straightforward. The Authors hesitate to submit nonstandard graphics without Editor's approval. However, this is a great idea for future work and the development of theoretical frameworks in this area.

• Reviewer 2 Comment 3: "Table 1 needs to be reorganized." To streamline Table 1, the portions of the table discussing race/ethnicity, educational background, and other demographics were removed as the information was already provided in the text and was, therefore, redundant. 

• Reviewer 2 Comment 4: "Most of the results revealed by the different regression approaches have r2 lower than 50%, the coefficients should be explained with discretion. If similar to the mainstream of the literature research, authors may need to explain this in their limitation sections." Effect sizes in social sciences are often much smaller than 50% (Gignac & Szodorai, 2016), and all r2 estimates were significantly different from zero, indicating that they explained a meaningful proportion of variance in outcomes.

o Gignac, G. E., & Szodorai, E. T. (2016). Effect size guidelines for individual differences researchers. Personality and individual differences, 102, 74-78. https://doi.org/10.1016/j.paid.2016.06.069

• Reviewer 2 Comment 5: "The odds should be reported as odds ratios. I don’t think Table 7’s results aligned with the main manuscript, for the odds of neurodevelopmental disorder and schizophrenia spectrum disorder. Please also check the other results. It would be better if authors could place the tables directly under the text where cited." In Table 7, rather than "Results reported are unstandardized," it now reads, "Results reported are unstandardized logistic regression coefficients." Odds ratios are reported, so the word, "ratios," was added for clarity. [INSERT TABLE X HERE] is noted for the Editors to place to tables under the text where cited (as we believe is standard for manuscripts submitted to PLOS One). 

• Reviewer 2 Comment 6: "How will the findings from this study help to improve the social environmental and legislation procedures as well as global cooperation to avoid the CSE? I think the authors should make their implications much clearer in the discussion section." A limitation mentioned in the Discussion section of this manuscript is that the investigation does not have a control group. In other words, all of the individuals represented in the study experienced CSE. Therefore, we cannot speak on how CSE can be avoided. Rather, we discuss the implications in terms of decreasing severity, if severity is defined by younger ages of entry and longer experiences of CSE. Social and legislative implications are listed in the Discussion section titled, "Barriers to Getting Out and Staying Out," and include universal healthcare, improved community-based mental health services, paid maternity leave, affordable childcare, vacatur, and alternative sentencing. In addition, since the data were collected from individuals located in the state of Georgia in the United States, the authors hesitate to speak to the global generalizability of the findings. Even still, it helps us to understand vulnerabilities within contexts, and future research should consider other global contexts.

• Regarding Ethics Statement: We have amended our Ethics statement to comply with PLOS ONE requirements to include information regarding waiver of consent. Thus, the ethics statement reads as follows: "Data were used with permission from Frontline Response International, Inc. All research activities were conducted under the oversight of Auburn University's Institutional Review Board (IRB) under protocol #22-192. This secondary analysis of existing data was granted a waiver of consent by Auburn University's IRB, and researchers accessed the data on April 27, 2022." This statement is also included in the Methods section. 

• Regarding Data Availability Statement: We have amended our Data Availability statement to comply with PLOS ONE requirements regarding confidential data to state, "This manuscript's data are not publicly available due to restrictions imposed by Frontline Response International, Inc. These restrictions are intended to protect client confidentiality. However, the data underlying the results presented in the study are available from Frontline Response International, Inc. at info@frontlineresponse.org. Data may be provided to eligible individuals upon reasonable request and with the completion of required prerequisites, such as a Data Use Agreement."

Again, we would like to thank the Reviewers for their time and thoughtful feedback, and we look forward to future correspondence on this matter. 

Sincerely,

The Authors

---

## [Editor Report · Decision Letter 1]

11 Sep 2024

Sex Trafficking Vulnerabilities in Context: An analysis of 1,264 case files of adult survivors of commercial sexual exploitation

PONE-D-24-06079R1

Dear Dr. Furlong,

We’re pleased to inform you that your manuscript has been judged scientifically suitable for publication and will be formally accepted for publication once it meets all outstanding technical requirements.

Kind regards,

Shivanand Kattimani

Academic Editor

PLOS ONE

Additional Editor Comments (optional):

I have read the comments from the reviewers and the authors replies.

Replies are satisfactory.

---

## [Editor Report · Acceptance letter]

22 Oct 2024

PONE-D-24-06079R1 

PLOS ONE

Dear Dr. Furlong, 

I'm pleased to inform you that your manuscript has been deemed suitable for publication in PLOS ONE. Congratulations! Your manuscript is now being handed over to our production team.

Kind regards, 

on behalf of

Dr. Shivanand Kattimani 

Academic Editor

PLOS ONE